

# Scoring and ranking probabilistic seismic hazard models: an application based on macroseismic intensity data

Vera D'Amico[1], Francesco Visini[1], Andrea Rovida[2], Warner Marzocchi[3], Carlo Meletti[1]

[1]Istituto Nazionale di Geofisica e Vulcanologia, Pisa, 56125, Italy
[2]Istituto Nazionale di Geofisica e Vulcanologia, Milan, 20133, Italy
[3]Department of Earth, Environmental, and Resources Sciences, University of Naples, Federico II, Naples, 80126, Italy

*Correspondence to*: Vera D'Amico (vera.damico@ingv.it)

**Abstract.** A probabilistic seismic hazard model consists of a set of weighted models/branches that describes the center, the body, and the range of seismic hazard. Owing to the intrinsic nature of this kind of analysis, the weight of each model/branch
represents its scientific credibility. However, practical uses of this model may sometimes require the selection of one or a few hazard curves that are sampled from the whole model, that often consists of thousands of branches. Here we put forward an innovative procedure that facilitates the scoring, ranking and selection of those hazard curves to account for the requirements of a specific application. The approach consists of a careful quality check of the data used for scoring and the adoption of a proper scoring rule. To show the applicability of this approach, we present an example that consists of scoring
and ranking a set of multiple models/branches constituting a recent seismic hazard model of Italy. To score these branches, hazard estimates produced by each of them are compared with time-series of macroseismic observations available in the Italian macroseismic database for a carefully selected set of localities deemed sufficiently representative, homogeneously distributed in space and complete with respect to time and intensity levels. The proper scoring parameter used for such a comparison is the logarithmic score, which can be always applied independently from the distribution of the data.

## 1 Introduction

Probabilistic Seismic Hazard Analysis (PSHA) is the most used scientific component to define an appropriate seismic building code. Owing to the important practical implications, PSHA models have to be widely accepted by a large scientific community. This acceptance is usually achieved by using commonly adopted procedures to calculate PSHA, and the full description of associated uncertainties is one of the key points in reliable models (Gerstenberger et al., 2020).

PSHA is usually built considering different models or branches of a logic tree, which mimics the so-called epistemic uncertainty, i.e., our ignorance of the true seismic hazard value. A critical aspect in describing quantitatively the distribution of the epistemic uncertainty is the way in which the weight of each model or branch is assigned.

Conceptually, the weighting of each model can follow two main general procedures (e.g., Albarello and D'Amico, 2015): the first one is ex-ante, that is by considering inherent properties of each competing PSHA model, i.e., its ability to take into
account the current knowledge of the underlying physical process evaluated by panels of experts; the second one is ex-post,



that is by empirically scoring a set of alternative models by comparing the forecasting performance of their outcomes with available seismic observations. The first approach was the most commonly adopted in the past (e.g., Stucchi et al., 2011; Woessner et al., 2015), whereas today, thanks to the large availability of seismic data for comparisons, state-of-the-art PSHA models tend to adopt a combination of the two approaches (e.g., Danciu et al., 2021; Petersen et al., 2023). For example, in the recent PSHA model for Italy called MPS19 (Meletti et al., 2021) the weight of each branch was assigned according to both ways, that is testing the performance of its components, i.e., seismicity and ground-motion attenuation models, against available observations and through the evaluation of the models by panels of experts. Worthy of note, independently from the specific scheme adopted, the weighting of each PSHA model is based on scientific considerations.

The use of a PSHA model for practical applications may require additional evaluations. Actually, most practical applications require the choice of one (or a few) hazard curves that are sampled from the model. For instance, many current building codes use arbitrarily the mean hazard, neglecting *de facto* the dispersion of all other hazard curves. Here we propose an innovative post-processing scoring strategy that facilitates the ranking and sampling of models/branches of a PSHA model to consider specific requests from stakeholders, e.g., those responsible for planning seismic risk reduction strategies.

We introduce the procedure through an application aimed to score and rank a set of multiple models/branches that constitute the MPS19 seismic hazard model of Italy according to their fit with macroseismic intensity data available in a large set of selected sites. The scoring procedure consists of a careful quality check of the data used for scoring and the adoption of a proper scoring rule.

MPS19 consists of 11 groups of seismicity models (each composed by a set of sub-models, for a total of 94 seismicity models) combined with three Ground Motion Models (GMMs) for the active shallow crustal areas (Bindi et al., 2011; Bindi et al., 2014; Cauzzi et al., 2015), with two GMMs for the subduction zone of the Calabrian Arc (Skarlatoudis et al., 2013; Abrahamson et al., 2016) and one for the volcanic areas (Lanzano and Luzi, 2020), producing a total of 564 branches (for more details on MPS19, see Visini et al., 2021 for seismicity models, Lanzano et al., 2020 for GMMs, and Meletti et al., 2021 for the whole model).

Specifically, the scoring procedure proposed here consists of comparing the hazard of each branch of MPS19 with the time-series of macroseismic observations ("seismic histories") available in the Italian macroseismic database DBMI15 v1.5 (Locati et al., 2016; https://emidius.mi.ingv.it/CPTI15-DBMI15_v1.5/query_place/) for a set of localities deemed sufficiently complete.

The proper scoring parameter for such a comparison is the logarithmic score (Gneiting and Raftery, 2007), which can be always applied independently from the specific distribution of the data; when the data follow a Poisson distribution, the logarithmic score is also named Log-Likelihood score (LL):

$$LL = \sum_{i=1}^{N} \log(p_i) \tag{1}$$





where $p_i$ is the probability of a given model for each of the $N$ observations. Gneiting and Raftery (2007) show that many other metrics, such as probability, do not have these characteristics and should not be used.

The main phases of the proposed procedure are the following: i) identification of the testing localities where to compare hazard estimates of the individual models with available seismic histories, ii) building of the datasets of observed and expected macroseismic intensities for each locality, iii) comparison between estimates from each branch and observed data in terms of LL of the differences between the number of macroseismic data predicted by the model and the number of those observed for different intensity degrees, and iv) scoring and ranking of the models.

Although our application is focused on a specific PSHA model, we emphasize the generalizability to any other model and kind of observations (e.g., accelerometric data, fragile geological structures), provided they are treated with *ad hoc* procedures.

## 2 Building the datasets of observed and expected intensities

The first step of our procedure is the identification of the set of localities for evaluating the consistency of PSHA models with available observations; then, for each site, two datasets of macroseismic intensities, one of observed data and one of intensities expected according to the hazard estimates, have to be built.

### 2.1 Selection of the testing localities

The selection of the sites where to compare PSHA models' outputs with available observations represents one of the most crucial issues of the scoring procedure and thus needs great attention.

In order to have a representative set of sites to perform tests, the selected localities have to guarantee: i) a geographical coverage as dense and uniform as possible throughout the whole investigated area, in relation to both high and low hazard regions, and ii) seismic histories with a significant number of data, spanning long time periods and covering a wide range of intensity values (see the examples in Fig. 1).




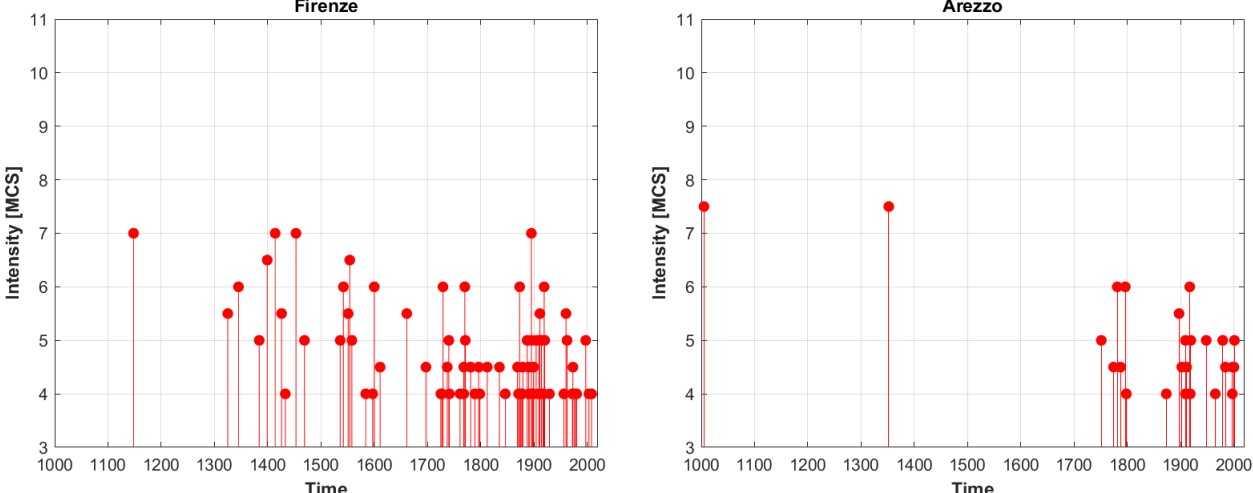

**Figure 1: Examples of seismic histories of two nearby Italian provincial capitals. The seismic history of Firenze (left) is extended and regular in time (for intensity larger than 4 MCS - Mercalli-Cancani-Sieberg scale; Sieberg, 1923), whereas the one of Arezzo (right) shows significant gaps in time (i.e., in the periods 1350-1750, and 1800-1850).**


In the application to the Italian territory described here, we first identify 133 sites corresponding to 97 provincial capitals and 36 localities selected trying to fulfil the above criteria.

We further check the representativeness of their seismic histories, provided by DBMI15 v1.5, comparing the seismic hazard estimates computed at each locality by means of the so-called "site" approach to PSHA (SASHA; D'Amico and Albarello,

2008) using: i) only the observed intensity data in DBMI15, and ii) the observed data integrated with "virtual" intensities calculated from earthquake parameters of the CPTI15 v1.5 catalogue (Rovida et al., 2016) through an intensity attenuation relationship (Pasolini et al., 2008, recalibrated by Lolli et al., 2019). High differences between the two resulting hazard estimates may indicate localities with "poor" seismic histories and/or with evident lack of data that should not be used for scoring.

On the basis of this analysis, which lead to eliminate or replace 13 localities that might bias the tests (six sites are retained to avoid large uncovered areas although they have poor seismic histories), and of the re-examination of the geographical distribution of the resulting sites, a further analysis is carried out to thin out very dense areas in Northern and Central Italy and to increase the density in some areas in the South. The final set of 124 locations selected for scoring is shown in Fig. 2.

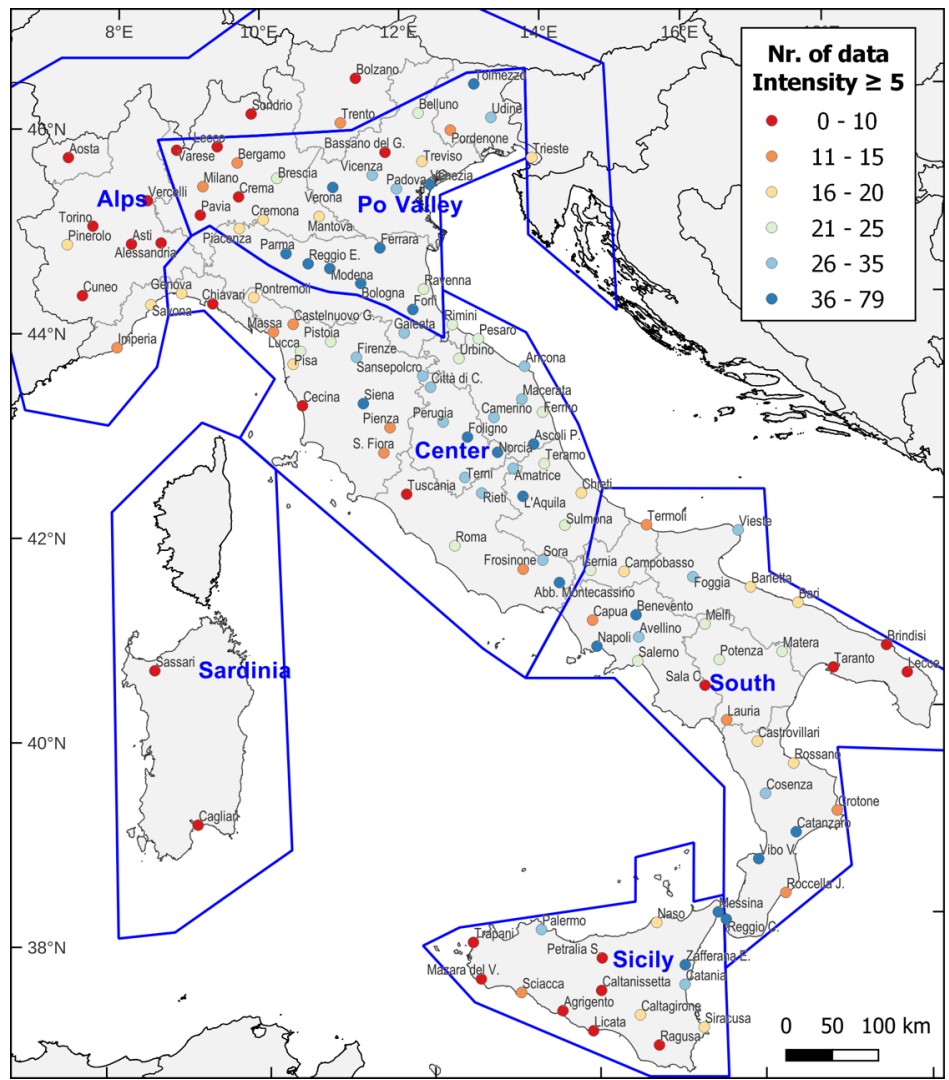

**Figure 2:** Map of the 124 localities selected for scoring showing the number of intensity data ≥ 5 MCS, for each of them. The polygons identify the six macro-areas used for subsequent tests.

## 2.2 Completeness periods of site seismic histories

In DBMI15, 9308 intensity data are referred to the selected localities and are associated to 2400 earthquakes spanning the period 1000-2014 and the whole range of intensity degrees, up to 10-11 MCS (Mercalli-Cancani-Sieberg scale; Sieberg, 1923).

However, the consistency check of the hazard estimates provided by a given model with the macroseismic observations available at the selected localities requires that the number of intensities expected from the model at each site is compared



with a complete set of observed intensities for each intensity degree. As a consequence, to calculate the number of macroseismic data, both observed and expected, at each site it is first necessary to estimate the completeness time intervals for each intensity degree, i.e., the periods in which it is reasonable to assume that all the earthquake effects above a given intensity have been actually reported in the seismic history (see Stucchi et al., 2004; Antonucci et al., 2023). For this reason, the completeness of the site seismic histories is different from the completeness of the earthquake catalogue. Indeed, the

effects of an earthquake occurred in the complete period of the catalogue might not be recorded at a given site for several reasons (e.g., they were not documented, documentation exists but has not been analyzed, and so on).

In our case study, the completeness time intervals for each site are defined using the statistical procedure of Albarello et al. (2001) applied to observed data with intensity greater than or equal to 5 MCS according to the following procedure:

– intensity data related to earthquakes in CPTI15 identified as "mainshocks", according to the declustering method used

in MPS19 (Gardner and Knopoff, 1974), are considered;

– only intensity data of earthquakes before 2006 are used, because after that year the systematic collection of macroseismic data ceased and DBMI15 is incomplete (Antonucci, 2022);

– intensities expressed in DBMI15 as non-numerical values, e.g., F for "felt", HD for "heavy damage", and so on (see Rovida et al., 2020 for their complete list), are discarded;

– uncertain intensities between adjacent integer degrees (e.g., 6-7 MCS) are treated as either the lowest degree (option 1) or the highest one (option 2).

For each Macroseismic Intensity (MI) threshold, two completeness estimates are therefore obtained, in terms of the starting year of the complete period ($T_c$), referred to the two options for assigning the uncertain degrees described above. To take the

uncertainty in the estimation of completeness into account, the two $T_c$ values corresponding to the median and the 75[th] percentile of the completeness function provided by the adopted method are considered, for a total of four $T_c$ values. The estimates of $T_c$ corresponding to the 25[th] percentile of the completeness function are not taken into account as they are considered unrealistic, especially for high degrees (see the example in Fig. 3). Finally, in case the completeness period of a given intensity threshold is shorter than that of the lower one (e.g., for degree 9 MCS in Fig. 3 on the right), the latter period

is considered for both the thresholds.




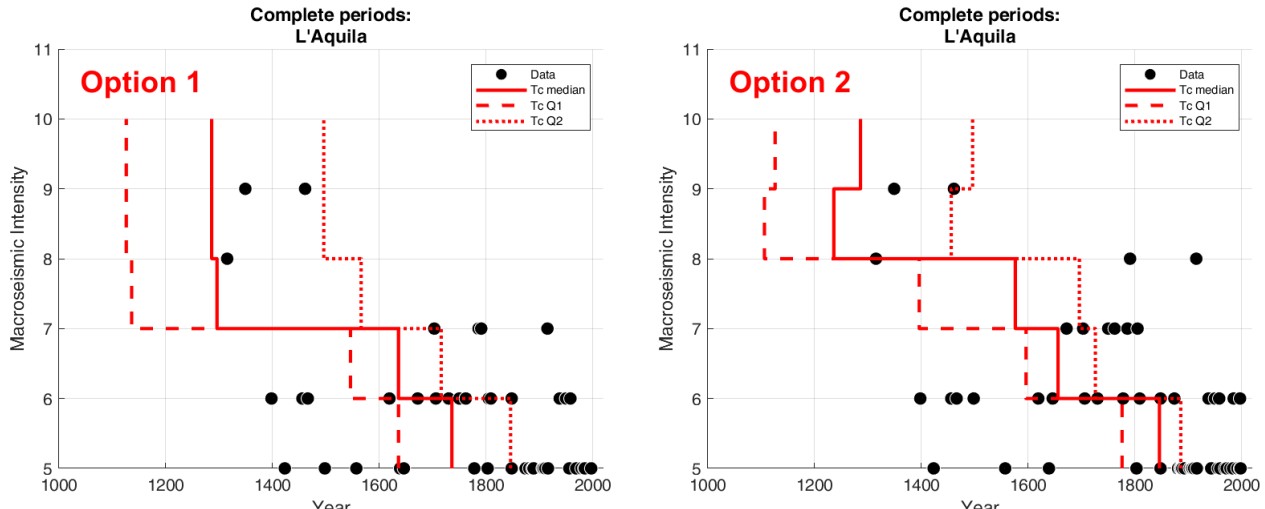

**Figure 3: Example of completeness graph for the city of L'Aquila according to the two options for assigning uncertain degrees: on the left, uncertain degrees are assigned to the lowest degree (option 1); on the right, to the highest degree (option 2). The red bars indicate, for each MI threshold, the three estimates of the completeness starting year $T_c$ (the median value with solid line, the 25th and 75th percentiles with dashed lines). Black dots correspond to intensities observed up to 2006, extracted from DBMI15.**

## 2.3 Dataset of observed intensities

According to the procedures described above, the dataset of observed macroseismic intensities for each testing locality is built counting, for each MI degree, the numbers of data after the two different completeness starting years $T_c$, i.e., those corresponding to the median value and the 75th percentile of the completeness function, considering both options 1 and 2 for treating the uncertain degrees.

Thus, four estimates of the number of observed data for each intensity degree are obtained, corresponding to:

    i)   option 1 and the median $T_c$ value (2100 data);

    ii)  option 1 and the 75th percentile $T_c$ value (1671 data);

    iii) option 2 and the median $T_c$ value (2557 data);

    iv) option 2 and the 75th percentile $T_c$ value (2076 data).

(In brackets, the total number of data referred to all intensity degrees and selected localities is reported).

The resulting numbers of data are finally cumulated to obtain the observed exceedances for each considered MI degree at each locality.





## 2.4 Dataset of expected intensities

The number of expected intensity data at each site on the basis of the hazard estimates provided by the individual branches of MPS19 is computed as follows:

– the hazard curve for each branch is calculated assuming a value of $V_{S,30}$ equal to 600 m/s, that is for EC8 soil category B instead of A ($V_{S,30}$ = 800 m/s) to which the MPS19 model refers. This is because macroseismic intensity values quantify the earthquake effects (in particular the levels of building damage) observed in extended localities that, in Italy, are generally located on class B soils ($360 \leq V_{S,30} < 800$ m/s, see e.g., Mori et al., 2020) rather than on rocky soils;

– from each hazard curve, expressed as annual rates of exceedance of different levels of shaking in terms of Peak Ground Acceleration (PGA) or Spectral Acceleration (SA), the corresponding annual rates of occurrence are obtained. These are then converted into occurrence rates of different degrees of intensity $\lambda$(MI) through the Ground Motion Intensity Conversion Equation (GMICE) by Gomez Capera et al. (2020), taking into account the associated uncertainties, as follows:


$$\lambda(\text{MI}) = \sum_{j=1}^{M} \lambda(x_j) P(\text{MI}|x_j) \tag{2}$$

where $\lambda(x_j)$ is the annual occurrence rate of each of the $M$ levels of PGA (or SA) in the hazard curve and $P(\text{MI}|x_j)$ corresponds to the conditional probability distribution of the GMICE, as proposed by D'Amico and Albarello (2008);

– the rates of occurrence in intensity estimated in this way are then multiplied by the lengths of the corresponding completeness periods to obtain the number of macroseismic data expected for each intensity degree. As done for the observed intensity data, the four estimates of the completeness periods are considered (starting from the median $T_c$ value and the 75th percentile of the completeness function and for the two options for assigning the uncertain degrees);

– the resulting numbers of data are finally cumulated to obtain the expected exceedances for each MI degree.

Figure 4 shows an example of the comparison between the number of observed and expected macroseismic data in the locality of Amatrice for different intensity thresholds.

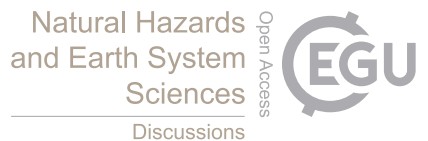
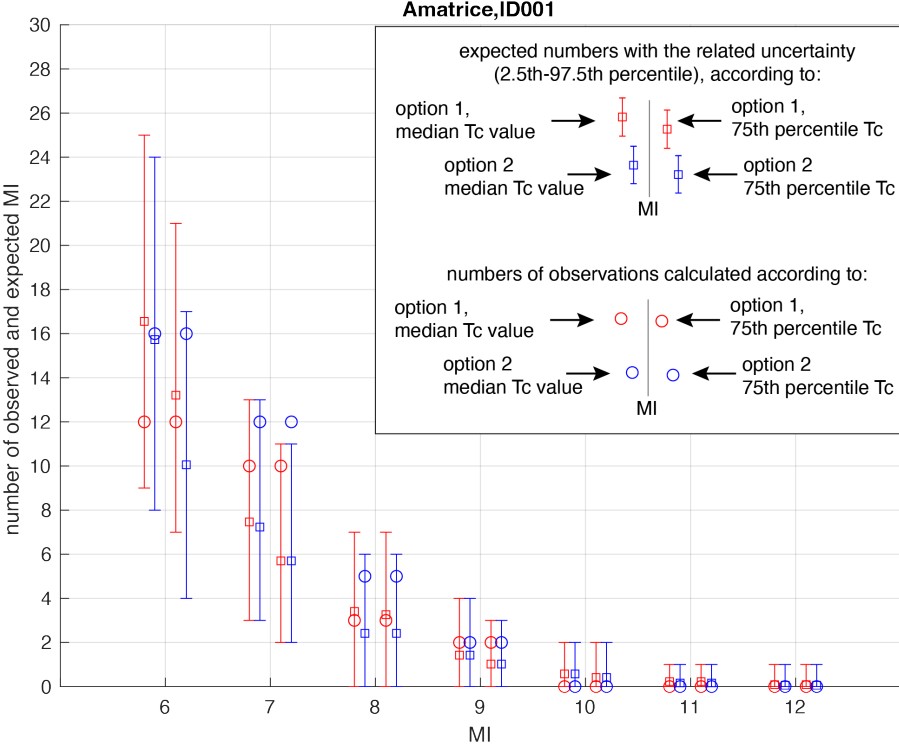


**Figure 4: Comparison, for different MI thresholds, between the number of observed macroseismic data in the locality of Amatrice and the number of expected intensities from one of the branches of MPS19 (ID001, calculated for soil class B).**

## 3 Consistency test between hazard estimates and macroseismic observations

The parameter used for evaluating the consistency of the predictions of a given PSHA model with the macroseismic observations available for the testing localities is the Log-Likelihood score LL (Eq. (1)). In this application, comparisons between forecasts and observations are made for individual branches (or models) of MPS19 starting from the hazard curves calculated at each testing site for soil class B ($V_{S,30}$ = 600 m/s) for PGA, SA 0.2 s and 1 s, that are considered the most relevant spectral periods for engineering purposes. The total number of analyzed branches is 282 out of the 564 of MPS19,

because the hazard values estimated at the testing sites using the two alternative GMMs selected for the subduction zone are almost identical and only the branches adopting the model of Skarlatoudis et al. (2013), that obtained the highest weight, are considered.





### 3.1 Calculation of the Log-Likelihood score (LL)

As described in the previous section, for each considered testing locality and for each PSHA branch, four pairs of observed
and expected numbers of intensity data are obtained for each MI threshold, corresponding to the four different estimates of
the completeness periods, i.e., for the median value and the 75[th] percentile of the completeness function and for the two
options for treating uncertain intensity degrees (see the example in Fig. 4).

For each site and branch, for each MI threshold and for each of the four pairs of observed and expected numbers of intensity
data, the probability $p$ of the tails of the Poisson distribution is calculated through the following algorithm (Zechar et al.,
210  2010):

– if the number of observed data ($N_{obs}$) is greater than the number of expected ones ($N_{exp}$):

$$p = 1 - F((N_{obs} - 1); \, N_{exp}) \tag{3}$$

– if the number of observed data is lower than or equal to the number of expected ones:

$$p = F(N_{obs}; \, N_{exp}) \tag{4}$$

where $F$ is the right-continuous Poisson cumulative distribution function with expectation $N_{exp}$ evaluated at $N_{obs}$:


$$F(N_{obs} | N_{exp}) = e^{-N_{exp}} \sum_{k=0}^{|N_{obs}|} \frac{N_{exp}^{k}}{k!} \tag{5}$$

The two probabilities ($p$), defined in Eqs. (3) and (4), respectively answer one of the following questions: is the forecast too
low (Eq. (3)) or too high (Eq. (4)) compared to the observations?
For each site, we then calculate the weighted average of the four logarithmic scores (LL in Eq. (1)), considering the four
pairs of observed and expected numbers of data for intensity greater than or equal to 6 (MI6+) and 8 (MI8+) MCS. These
intensity levels correspond to the threshold of slight and structural building damage, respectively. The weighted average of
observed and expected data is calculated by equally weighting the two estimates obtained from the median value and the 75[th]
percentile of the completeness function and attributing different weights to the two options for treating uncertain degrees as
follows: i) 0.75 to option 1 (i.e., uncertain degree assigned to the lower MI value), and ii) 0.25 to option 2 (i.e., uncertain
degree assigned to the higher MI value). This choice is consistent with the definition of uncertain intensity values (Grünthal,
1998).

The LL value calculated in this way is defined as LL$_{site}$.



### 3.2 Estimates of LL for each model

To identify the models that produce the hazard estimates most consistent with the macroseismic observations at the testing sites, we initially calculate, for each branch, the sum of the $LL_{site}$ values relating to the 124 selected localities, defined as $LL_{sum}$, for the three spectral periods (PGA, SA 0.2 s and 1 s) and the two intensity thresholds (MI6+ and MI8+) considered. Figure 5 shows the $LL_{sum}$ values for PGA for all branches: the smaller (closer to zero) is the value, the higher is the consistency between the model's outcomes and the observations.


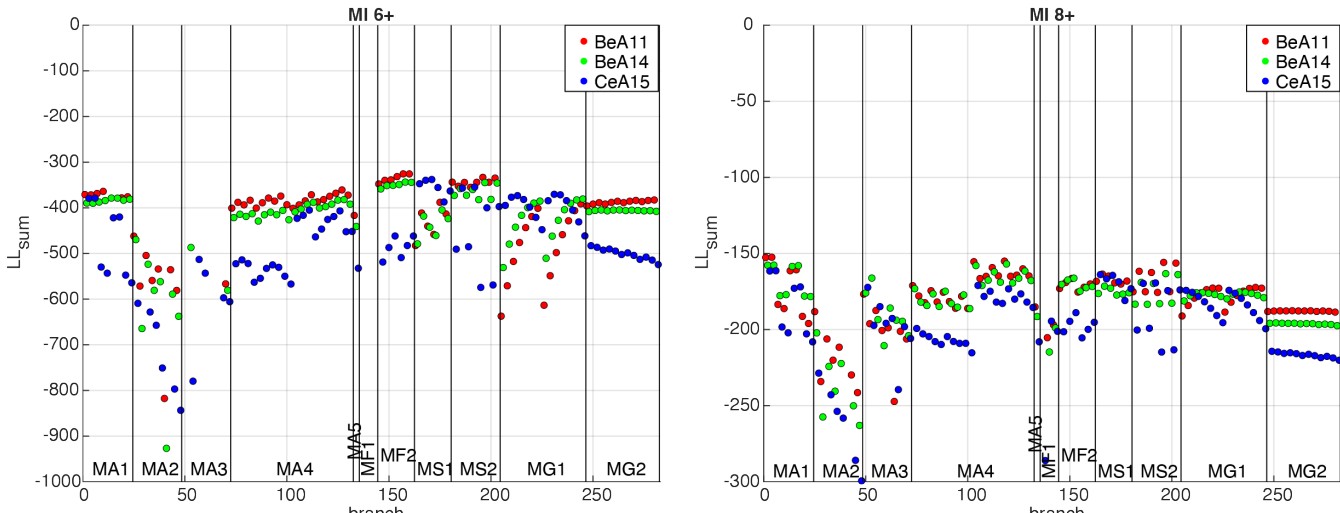

**Figure 5: Values of $LL_{sum}$ (sum of $LL_{site}$ of all localities) calculated for each of the 282 considered branches of MPS19 for PGA, MI6+ (left) and MI8+ (right). The branches are represented in abscissa from left to right grouped according to the 11 seismicity models (for the description of models, see Meletti et al., 2021; Visini et al., 2021), and colored according to the three GMMs**
**adopted for active crustal areas, namely: "BeA11" (Bindi et al., 2011), "BeA14" (Bindi et al., 2014), "CeA15" (Cauzzi et al., 2015). Note that the y-axis for MI6+ is truncated for the purpose of visualization, as a few values tend toward negative infinity.**

Then, to test the performance of each branch over different regions of the Italian territory, we group the selected localities according to the six macro-areas defined in MPS19 to estimate the completeness of the CPTI15 catalogue, that is: Alps, Po
Valley, Center, South, Sardinia, Sicily (see Fig. 2). Since these macro-areas include different numbers of sites, the average (instead of the sum) of the $LL_{site}$ values is calculated for both the entire set of 124 localities and for the localities in each macro-area (Sardinia is excluded since it has only two testing sites). Therefore, six $LL_{mean}$ values are obtained for each branch. Figure 6 shows the resulting $LL_{mean}$ values for PGA, for the two intensity thresholds MI6+ and MI8+ and for each adopted GMM.






Figure 6: Values of LL$_{mean}$ for each considered branch of MPS19 for the localities in five macro-areas and for all the sites, for PGA, MI6+ (left) and MI8+ (right), and for each adopted GMM. The branches are represented in abscissa from left to right grouped according to the 11 seismicity models.





As shown, almost all the branches seem to give a good agreement (i.e., $LL_{mean}$ values closer to zero) in the Alps macro-area,
characterized by a much smaller number of sites. In the other four macro-areas, that are more significant in terms of the
number of sites, the results appear to be very different depending on the different seismicity models. In particular, for the
BeA11 and BeA14 GMMs, some groups of branches (e.g., MA2, MA3, MF1, MG1, MG2) show generally poorer
performance in terms of $LL_{mean}$ values and a considerable geographical scatter, whereas others (e.g., MA1, MA4, MF2,
MS2) show values of $LL_{mean}$ that are generally smaller and more stable in the four macro-areas. The plots of $LL_{mean}$ values
for SA 0.2 s and 1 s are reported in the Supplement (Fig. S1 and S2).

In order to evaluate the stability of the performance of each branch in the different areas, we then calculate the dispersion of
the $LL_{mean}$ values among the four macro-areas including the highest number of localities (Po Valley, Center, South, Sicily) as
the width of the interval between the 2.5th and 97.5th percentiles. The percentiles are estimated using a non-parametric
distribution of the four $LL_{mean}$ values. Obviously, a different choice of the distribution might lead to changes in the
percentiles, but the aim is only to give an order of magnitude of the dispersion among the measures in the macro-areas.
Figure 7 shows the dispersion values computed for PGA, for the two intensity thresholds considered (plots for SA 0.2 s and
1 s are displayed in Fig. S3 of the Supplement).

The $LL_{mean}$ values computed using the entire set of localities and the relative geographical dispersion are then used to
establish a ranking of the branches.

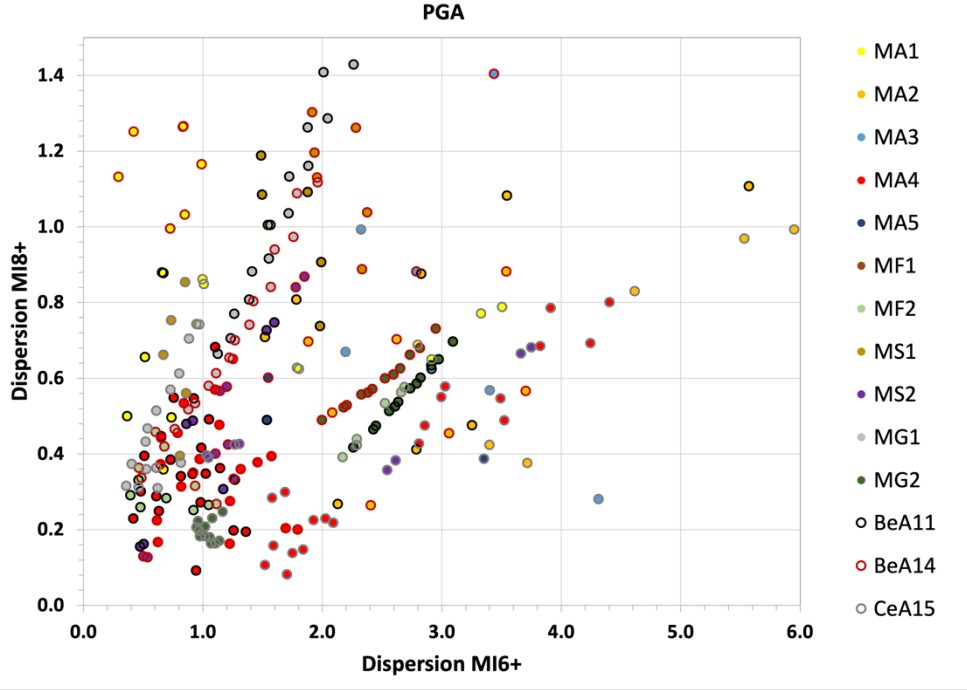

**Figure 7: Dispersion of the $LL_{mean}$ values among the four more representative macro-areas for each branch, for PGA, MI6+ and MI8+. The color of the dots indicates the seismicity model, while the color of the borders indicates the GMM used in that branch.**





## 3.3 Ranking of the models

Following the described procedure, a LL$_{mean}$ value for MI6+ and MI8+, for the three spectral periods (i.e., PGA, SA 0.2 s
and 1 s), is assigned to each of the 282 considered branches of MPS19, as well as an estimate of the dispersion of these
values among the four more representative macro-areas.

For each spectral period, the branches are then ranked according to the values of LL$_{mean}$ and the relative geographical
dispersion, assigning the 1$^{st}$ place to the branch with the "best" result (LL$_{mean}$ value or dispersion closest to zero) and the
282$^{nd}$ place to the "worst" one.

Initially, comparison plots of the ranks based on LL$_{mean}$ values for the two MI thresholds are produced, focusing the attention
on those branches that fall within the 10$^{th}$ percentile. This choice, however subjective, result too restrictive, since none of the
branches fall within this range for all the considered spectral periods. It is then decided to expand the selection criterion.
Taking into account the first 70 positions (corresponding to the first quartile) in the LL$_{mean}$ tests, for both intensity thresholds
and for the three spectral periods, we select 35 branches for PGA and 37 branches for SA 0.2 s and 1 s, representing all the
GMMs used and different seismicity models of MPS19.

Figure 8 shows the placement of each branch in the LL$_{mean}$ test for PGA; the plots for SA 0.2 s and 1 s are reported in the
Supplement (Fig. S4). In all the plots, the best ranks are generally occupied by the same models, for both MI thresholds.

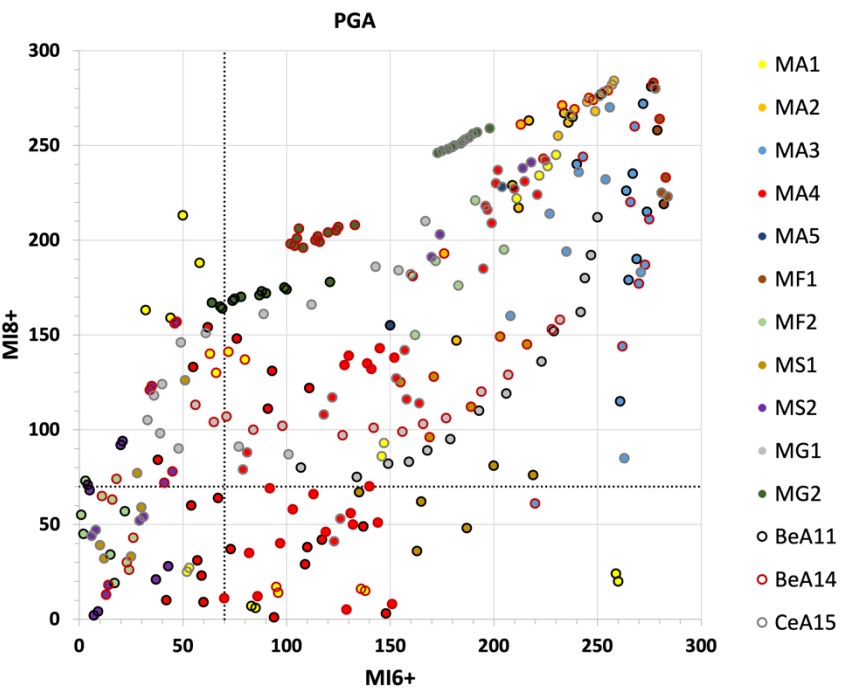

**Figure 8: Comparison between the ranking of the branches for PGA, based on the LL$_{mean}$ values for MI6+ and MI8+. The color of**
**the dots indicates the seismicity model, while the color of the borders indicates the GMM used in that branch. Black dotted lines**
**identify the 70$^{th}$ position in both rankings.**





The geographical dispersion of $LL_{mean}$ values among the four more representative macro-areas is then considered. For each spectral period, a ranking of the branches is made also according to this parameter and the ranks are grouped in three classes, for both MI6+ and MI8+, that is: i) rank ≤100, ii) rank 101-150, and iii) rank >150.

An overall rank (from 1, best rank, to 5) is then assigned to each selected branch, for each of the three spectral periods, based on the ranking class for the two intensity thresholds (see the abacus in Table 1).

**Table 1: Abacus built to assign an overall rank to each selected branch, for each spectral period considered, on the basis of its rank resulting from the dispersion of $LL_{mean}$ values in the four macro-areas.**

|  |  | Dispersion rank MI8+ | | |
|---|---|---|---|---|
|  |  | 1-100 | 101-150 | >150 |
| **Dispersion rank MI6+** | **1-100** | 1 | 2 | 3 |
|  | **101-150** | 2 | 3 | 4 |
|  | **>150** | 3 | 4 | 5 |


The overall rank could allow practitioners to further sample and/or (re-)weight the various branches according to the practical constraints of a specific application. Table 2 shows the overall ranks for the 20 best performing selected branches for PGA, SA 0.2 s and 1 s. As an example, if one decides to consider those models with an overall rank equal to 1 or 2 for all

the spectral periods, only the first six should be selected.





**Table 2: Overall ranks of the 20 best performing selected branches, according to the abacus in Table 1. The following columns report the position in every computed rank; the cells with gray background mark the branches falling in the first 70 positions in the LL$_{mean}$ ranking.**

| id | Seismicity model | GMM | PGA | 0.2 s | 1 s | PGA Rank_LL_MI6+ | Rank_LL_MI8+ | Rank_Disp_MI6+ | Rank_Disp_MI8+ | SA 0.2 s Rank_LL_MI6+ | Rank_LL_MI8+ | Rank_Disp_MI6+ | Rank_Disp_MI8+ | SA 1 s Rank_LL_MI6+ | Rank_LL_MI8+ | Rank_Disp_MI6+ | Rank_Disp_MI8+ |
|---|---|---|---|---|---|---|---|---|---|---|---|---|---|---|---|---|---|
| ID255 | MA4 | BeA14 | 1 | 1 | 1 | 70 | 11 | 53 | 53 | 59 | 17 | 85 | 3 | 22 | 16 | 3 | 71 |
| ID303 | MF2 | BeA14 | 1 | 1 | 2 | 24 | 26 | 38 | 93 | 21 | 40 | 88 | 27 | 7 | 52 | 54 | 117 |
| ID297 | MF2 | BeA14 | 1 | 2 | 1 | 23 | 30 | 70 | 54 | 27 | 50 | 123 | 9 | 4 | 46 | 80 | 84 |
| ID291 | MF2 | BeA14 | 1 | 2 | 2 | 26 | 43 | 100 | 39 | 42 | 70 | 146 | 4 | 1 | 44 | 123 | 61 |
| ID391 | MS2 | BeA11 | 2 | 1 | 2 | 7 | 2 | 61 | 117 | 1 | 29 | 31 | 63 | 32 | 51 | 134 | 94 |
| ID403 | MS2 | BeA11 | 2 | 1 | 2 | 9 | 4 | 65 | 119 | 2 | 26 | 34 | 67 | 24 | 47 | 126 | 89 |
| ID393 | MS2 | BeA14 | 3 | 1 | 2 | 13 | 13 | 104 | 149 | 8 | 6 | 51 | 90 | 14 | 15 | 53 | 133 |
| ID395 | MS2 | CeA15 | 2 | 3 | 2 | 31 | 54 | 121 | 98 | 25 | 37 | 117 | 148 | 6 | 18 | 83 | 136 |
| ID407 | MS2 | CeA15 | 2 | 3 | 2 | 29 | 52 | 119 | 97 | 23 | 33 | 111 | 146 | 8 | 26 | 84 | 141 |
| ID405 | MS2 | BeA14 | 4 | 1 | 2 | 14 | 18 | 108 | 157 | 9 | 7 | 61 | 94 | 18 | 17 | 56 | 137 |
| ID367 | MS2 | BeA11 | 4 | 2 | 1 | 37 | 21 | 134 | 198 | 5 | 21 | 106 | 73 | 3 | 30 | 78 | 99 |
| ID379 | MS2 | BeA11 | 4 | 2 | 2 | 43 | 28 | 144 | 205 | 6 | 22 | 115 | 81 | 2 | 31 | 75 | 108 |
| ID249 | MA4 | BeA14 | | 1 | 1 | 82 | 35 | 31 | 71 | 55 | 34 | 24 | 17 | 48 | 42 | 34 | 96 |
| ID371 | MS2 | CeA15 | 1 | 2 | | 8 | 47 | 88 | 81 | 10 | 27 | 84 | 131 | 59 | 154 | 107 | 242 |
| ID383 | MS2 | CeA15 | 1 | 2 | | 6 | 44 | 87 | 86 | 7 | 25 | 82 | 133 | 68 | 175 | 117 | 245 |
| ID251 | MA4 | CeA15 | | 2 | 2 | 81 | 88 | 151 | 6 | 62 | 55 | 124 | 7 | 30 | 27 | 8 | 145 |
| ID175 | MA4 | BeA11 | 1 | | | 67 | 64 | 71 | 2 | 96 | 178 | 122 | 145 | 229 | 266 | 253 | 182 |
| ID223 | MA4 | BeA11 | 1 | | | 60 | 9 | 42 | 77 | 36 | 89 | 67 | 49 | 155 | 181 | 169 | 37 |
| ID247 | MA4 | BeA11 | 1 | | | 57 | 31 | 13 | 49 | 32 | 113 | 12 | 91 | 137 | 200 | 138 | 27 |
| ID253 | MA4 | BeA11 | 1 | | | 42 | 10 | 25 | 46 | 30 | 101 | 52 | 72 | 168 | 210 | 180 | 41 |


## 4 Discussion and conclusions

We have introduced a new scoring strategy that may be used to rank and sample the multiple models/branches of a PSHA model. Scoring is inherently different from testing: the former term indicates approaches devoted at ranking and eventually

weighting a set of competing models, whereas testing procedures aim at evaluating the absolute predictive accuracy of each model, indicating if its outcomes are/are not compatible with observations to a given significance level threshold. Therefore, testing can allow to identify possibly wrong PSHA models, whereas scoring is aimed to compare models according to a specific metric of interest.

For the sake of example, we have scored and ranked alternative branches of the MPS19 seismic hazard model of Italy

(Meletti et al., 2021) according to their fit with long-term macroseismic intensity data available in a large set of sites. To





properly compare the performance of the different models/branches, a Log-Likelihood score has been assigned to each of them based on the comparison between numbers of expected and observed intensity data at each site for different shaking levels and spectral periods, not considering single return periods but the entire hazard curve.

In countries such as Italy, where the historical record is hundreds-of-years long, i.e., much more than the instrumental one, and macroseismic information covers the whole territory (Locati et al., 2022), the use of macroseismic intensity observations for scoring PSHA models could be more suitable than accelerometric recordings to consider the effects of earthquakes with large magnitudes and long return periods.

Of course, comparing PSHA outcomes in terms of PGA or SA with macroseismic data requires caution due to the use of GMICEs, that are empirical conversion relationships characterized by large uncertainties to be taken into account. In fact, if one simply converts the ground motion value (e.g., PGA) resulting by a PSHA model into macroseismic intensity just using the average estimates and discarding associated variance, the comparison could be severely biased. In the scoring procedure presented here, this issue is solved through the convolution of the relevant probability distributions (i.e., hazard curves and GMICE), as proposed by D'Amico and Albarello (2008). Moreover, the procedure takes into account both the peculiar nature of intensity values (discrete, ordinal, range-limited) and associated uncertainties (uncertain intensity values between adjacent integer degrees, completeness of site seismic history, etc.).

A further crucial issue related to using macroseismic intensity data for empirical scoring concerns the selection of sites where to compare PSHA models' outputs with available observations. In fact, to have a representative set of localities to perform tests, selected sites have to guarantee a geographical coverage as dense and uniform as possible throughout the study area (for both high and low hazard regions) as well as a significant number of macroseismic data at each site, covering long time periods and a wide range of intensity values. This clearly limits the use of macroseismic data as observables to those countries with long records of documentary information about the effects of past earthquakes at a sufficient number of sites (e.g., Fäh et al., 2011 for Switzerland; BRGM-EDF-IRSN/SisFrance, 2017 for France).

The presented procedure can be applied to any kind of model and set of observational data, for instance to rank and select branches of a complex PSHA model to get one outcome that better satisfies specific stakeholders' needs. In this regard, it is important to remark that our approach is based on a rigorous and quantitative procedure, although the definition of the thresholds and ranks for selecting branches is a subjective choice that depends on specific considerations and aims.

**Data availability**

CPTI15 v1.5 is available at https://doi.org/10.6092/ingv.it-cpti15
DBMI15 v1.5 is available at https://doi.org/10.6092/ingv.it-dbmi15



## Author contribution

All authors designed the scoring strategy and prepared the manuscript. VD wrote the draft of the paper and contributed to the selection of testing sites, the estimation of expected intensities and the ranking of the models. FV computed the completeness periods, the number of observed/expected intensity data and the LL values. AR performed the selection of testing sites and contributed to the estimation of completeness periods. WM proposed the scoring rule and CM performed the ranking of the models.

## Competing interests

The authors declare that they have no conflict of interest.

## Acknowledgements

This study has benefited from funding provided by the Italian Presidenza del Consiglio dei Ministri – Dipartimento della Protezione Civile (DPC) in the framework of the DPC-INGV Agreement B1 (2020-2021). This paper does not represent DPC official opinion and policies.

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
