# Peer review of "Scoring and ranking probabilistic seismic hazard models: an application based on macroseismic intensity data"

_Natural Hazards and Earth System Sciences, 2023_

## Author Response (AR1)

Dear Referee 1,

thank you for your review. We modified the manuscript following your comments and suggestions according to the answers reported below.

**REF:** 1. line 21. It is stated that PSHA is a "component to define an appropriate seismic building code". This is not exact. Actually, appropriateness of the seismic code relies on its capability to define safe constructung rules as a function of the expected ground shaking. The contribution of PSHA is defining where these rules should be applied and not defining the rules. Possibly it could be better changing the statement as "PSHA provide basic information for the proper application of the building code".

**ANSW:** We agree with your comment. The text was modified as suggested.

**REF:** 2. Line 38 (and line 21). The use of the term "scientific" should be clarified: what do Authors intend? Possibly different definitions could be adopted and the boundary between "scientific" and "not scientific" elements could be rather fuzzy. I kindly suggest to avoid this term.

**ANSW:** By the term "scientific" at line 38 we intended to distinguish between the analyses and evaluations made during the building phase of a PSHA model, that are based on available scientific knowledge and tests (e.g., about seismogenic process, ground motion modelling, etc.), and the further technical/political considerations and choices that are commonly required for practical application of PSHA outcomes to the building code and/or to seismic risk evaluations, such as the selection of one or a few hazard curves that are sampled from the whole model. To better explain this point, we modified the sentence as follows: "Worthy of note, independently from the specific scheme adopted, the weighting of each PSHA model relies on available scientific knowledge". The term "scientific" at line 21 was instead removed (see previous answer).

**REF:** 3. Line 64. the statement "pi is the probability of a given model for each of the N observations" should be reformulated as "pi is the probability that each model attributes to the i-th observation obove the N available".

**ANSW:** We corrected the text as suggested.

Dear Referee 2,

thank you for your review. We modified the manuscript following your comments and suggestions according to the answers reported below.

**REF:** • Lines 170-171: since hazard curves in terms of MI can be derived (there is some literature dealing with the hazard assessment in terms of macroseismic intensity for Italy), why not considering them in lieu of those in terms of Sa? This would avoid the use of conversion relationships which are affected by uncertainty.

**ANSW:** We agree that the most direct and correct approach would be comparing macroseismic observations with hazard estimates computed in terms of MI, avoiding the large uncertainty associated to conversion relationships. However, national PSHA models (including the MPS19 model considered in our paper) developed worldwide for building code applications are currently provided in terms of SA estimates, that are the shaking parameters required for seismic design definition. A sentence was added in the Introduction (at line 51 of the preprint) to describe the output parameters of MPS19.

**REF:** • As I said, the topic is of relevant interest. Notwithstanding, who, among those that are not scientists, can be interested in scoring PSHA models? A comment on that would be useful.

**ANSW:** The proposed scoring strategy could facilitate the ranking and sampling of models/branches of a complex PSHA model to consider specific requests from stakeholders, such as those responsible for planning seismic risk reduction strategies. In fact, most practical applications of PSHA (e.g., to the building code) require the choice of one or a few hazard curves that are sampled from the model (see lines 39-43 of the preprint). In this regard, the application presented in our paper aims at selecting the models/branches that minimize the difference between PSHA outcomes and macroseismic observations at a set of sites. A sentence to better clarify this point was added in the Introduction and in the Discussion and conclusions (at lines 46 and 325 of the preprint, respectively).